# Prevalence and determinants of hypertension among older adults: A comparative analysis of the 6th and 8th national health surveys of Bangladesh

**Probir Kumar Ghosh**[1]*, **Md. Golam Dostogir Harun**[1], **Ireen Sultana Shanta**[1], **Ausraful Islam**[1], **Kaniz Khatun E. Jannat**[2], **Haider Mannan**[3]

1 International Centre for Diarrhoeal Disease Research, Bangladesh (icddr,b), Dhaka, Bangladesh,
2 Doctoral student, School of Health Sciences, Western Sydney University, Sydney, NSW, Australia,
3 Translation Health Research Institute, Western Sydney University, Campbelltown Campus, Sydney, NSW, Australia

* probir@icddrb.org

**Data Availability Statement:** To access the DHS datasets used in my study, others can visit the DHS Program website and create an account. Once

## Abstract

### Background

Hypertension is a major public health concern in low-and middle-income countries. A nationwide Health, Population, and Nutrition Sector Development Program in Bangladesh has been shown to be effective in resource-poor settings. This article aims to investigate whether the prevalence and determinants of adult hypertension changed from 2011 to 2018.

### Methods

The determinants of adult hypertension were assessed in 2011 and 2018 data of Bangladesh Demographic and Health surveys. These two surveys included both men and women over the age of 34 years and measured their blood pressure, weight, height, and other covariates. For both surveys, we estimated the age-standard prevalence of hypertension and relative, attributable and mediated risk of determinants of hypertension using hierarchical mixed-effects sequential Poisson regression models.

### Results

The prevalence of adult hypertension increased by 10.9% from 29.5% in 2011 to 40.4% in 2018. The nationwide awareness program on the Health, Population and Nutrition Sector changed the risks associated with hypertension determinants over the years. During 2011, Socio-economic status (SES) was a major distal determinant of adult hypertension, explaining 21% of population-attributable risk (ART). However, other factors accounted for 90% of risk, mainly by excessive body weight (51%) and awareness of hypertension (39%). In contrast, SES only explained 16% of ART risk, with 97% of the risk mediated by excessive body weight (55%) and awareness of hypertension (41%).

logged in, they can browse the available datasets and download the ones they need. Here is a direct hyperlink to the DHS Program website: https://dhsprogram.com/. Here is a direct hyperlink to the page where you can browse and download DHS datasets: https://dhsprogram.com/data/available-datasets.cfm, Others can access the DHS datasets in the same manner as the authors. The DHS Program website is publicly accessible and anyone can create an account to download the datasets. The authors did not have any special access privileges that others would not have. The authors used the same public website and account creation process to download the DHS datasets as anyone else.

**Funding:** The authors received no specific funding for this work.The funders had no role in study design, data collection and analysis, decision to publish, or preparation of the manuscript.

**Competing interests:** The authors have declared that no competing interests exist.

## Conclusion

The study results highlight that hypertension among older adult was significantly increased over the six-year period. Specially, the socio-economic status, awareness of hypertension and excessive body weight were the significant determinants. Being awareness of hypertension and excessive body weight changed the causal pathways of socio-economic status. The results also highlight the value of studying the effect of non-communicable disease awareness programs to enhance our comprehension of factors influencing health.

## Introduction

Hypertension is one of the major public health problems, affecting an estimated 1.28 billion adult population globally. Most of these people live in low-middle income countries[1]. Hypertension is a prominent global health challenges, mainly in the older population threatening their ability to reduce cardiovascular disease (CVD) and stroke-related premature mortality and disabilities [2,3] and enormous economic consequences. [4]. The causal effects of hypertension are mainly experienced in the low-and middle income countries because of their nutrition status [5], demographic [6], and epidemiological transitions [7,8]. Furthermore, epidemiological studies in various low-and middle income countries have shown that non-communicable diseases, particularly hypertension, overweight, obesity, and diabetes mellitus, have a complex inter-relationship [9–12]. Additionally, numerous determinants affect non-communication diseases, with the well established recognition of socioeconomic status (SES) playing a key role [9,13].

It is widely recognized that a range of factors influences the likelihood of developing hypertension, of which the socio-economic status (SES) factor is mediated by the factors that are very close [14], and hierarchical modellings are primarily employed to adjust for confounder effect and sequentially identify risk factors [14–17]. For instance, SES continues to be a significant factor influencing hypertension [18], and is strong link to behavioral and economic factors such as, variation of physical activity, intake of foods, vegetables, fruits, high-fat dairy products and overall lifestyle. This is quite evident because differences between rich and less affluent manifest like social, food habit, lifestyle, and health system [19]. Adult people in the wealthy communities often have more health problems because they lead inactive and unhealthy lives [20].

The priority of Health Population and Nutrition Sector Development Program, Bangladesh (HPNSDP) was to increase awareness of non-communicable diseases (NCDs) and improve healthcare access to all population, particularly the poor, due to increase life expectancy at birth, reductions of malnutrition and micro-nutrient deficiencies and overall improvement of health [21]. This program provided direct effective way to improve adult health in Bangladesh. However, one of the main challenges was to reduce non-communicable diseases, especially hypertension in Bangladesh.

Several studies have shown the critical role of other non-communicable diseases, including excessive body weight and diabetes mellitus in reducing hypertension. They have shown that normal weight and non-diabetes mellitus can prevent the risk of hypertension significantly [22–24]. A few studies have evaluated the hierarchical effect of distal factors such as SES and proximal determinants on adult hypertension risk in Bangladesh. Recently, the 6th and 8th Bangladesh Demographic, and Health survey, two cross-sectional national-wide population-

based surveys in 2011 and 2017–18 have shown the prevalence of non-communicable diseases, awareness and controlling of hypertension. A non-robust methodology was used to measure the adult health indicators in 2011 and 2018, so the results were affected by several biases. The study showed a rise in hypertension prevalence of more than 10% among adult population older than 34 years within the two-survey time period. The survey reports also described the measurement of hypertension, self-awareness and control among individuals [25].

Some well-conducted quantitative risk assessments in non-communicable diseases are important to understand the risk factors for health [26–28]. These assessments helped to identify the preventive measures to reduce adverse health outcomes. A few critical factors directly contributed to hypertension by soaring exposure to immediate risk factors, and in turn, indirectly influencing other determinants [27]. It is likely that these factors act not only on hypertension, but also modify the pathways to hypertension risk. In this study, we assessed the prevalence of hypertension and how the magnitude of relative, attributable, and mediating risk of different determinants of hypertension changed among older adults in Bangladesh between 2011 to 2018.

## Methods

Bangladesh, a low- and middle-income country in South Asian population has around 18 million common health issues including non-communicable diseases. Sedentary and detrimental lifestyle including tobacco consumption, substance abuse is common; poor mental and physical health, injury, environmental quality, lack of immunization and healthcare access are also prevalent. The Health Population and Nutrition Sector Development Program, Bangladesh (HPNSDP) has been prioritized since 2011 to increase life expectancy at birth, reduce non-communicable diseases, and malnutrition [29,30]. The 6th Bangladesh Demographic and Health Survey in 2011(BDHS 2011) involved adult NCD indicators among those older than 34 years as a baseline for measuring the NCD. The preliminary results from BDHS 2011 revealed that one of three adults more than 34 years were hypertensive [25]. In order to have more focus on non-communicable diseases (NCD), the 4th HPNSP 2016–2021 included a component focusing on increasing life expectancy at birth, immunization coverage, reduction of malnutrition along with controlling the epidemiological shift and the double challenge of communicable and non-communicable diseases (NCD) in Bangladesh, with details provided elsewhere [31]. To assess adult health outcomes, the 8th Bangladesh Demographic and Health Survey (BDHS) was conducted in 2017–18 ((BDHS 2017–18)) which was the second survey to collect data on NCD measurements [25]. The children's health and nutrition key outcomes have been achieved, but adult population health outcomes are not under control. Each survey collected data on NCD among those aged 34 years and older (both women and men) from each third of the households in the survey. The survey involved 18,000 and 20,160 households obtained by using two-stage stratified cluster sampling method in 2011 and 2017–18 respectively. In the first stage, 675 clusters were selected from rural and urban areas in each of eight divisions. A sample of 30 households were systematically selected from each cluster in the second stage. A similar methodology and sampling strategy were used in both surveys which is described elsewhere [25,32]. To understand the adult hypertensive changes between 2011 and 2018, we used data of the 6th and 8th Bangladesh Demographic and Health surveys, (BDHSs) data conducted from March to October 2011 and from October 2017 to March 2018 respectively.

NCD related biomarkers and relevant information were collected in both surveys. Blood pressure (BP) was taken for all adults older than 34 years in both surveys using a digital oscillometric blood pressure measuring device (Model: UA 705, A&D company, origin of Japan) with automatic upper-arm inflation and an automatic pressure release by health workers during the

interview. The prevalence of hypertension was our outcome of interest. These surveys reported the systolic and diastolic BP measured three times in millimetres of mercury [mmHg] with at least a 5-minute interval between measurements, and then averaged. We classified an individual as hypertensive if he/she had systolic BP $\geq$ 140 mmHg and/or a diastolic BP $\geq$ 90 mmHg as per current National Guidelines for Management of Hypertension in Bangladesh [33].

In addition, at both surveys, individual and household characteristics were used in the study to assess confounders. These added SES, household assets and education, nutritional status related variables (height in meter and weight in kilogram). Participant's height in meter and weight in kilogram were measured by using standardized instruments and procedures, as detailed in the BDHS 2017–18 report [23,25]. A composite score of body mass index (BMI) as adult nutritional status for each adult was calculated by dividing weight in kilograms by height in meters squared ($kg/m^2$). We categorized BMI into four distinct groups; underweight (BMI<18.5), normal weight (BMI between 18.5 and 25), overweight (BMI between 25, 30), and obese (BMI above 30) [34,35].

## Statistical analysis

First, we used descriptive statistics to calculate the distribution of covariates in 2011 and 2018. Then, we used step-by-step bivariate analysis along with mixed effect Poisson regression models to identify potential risk factors associated with hypertension in adults. We calculated prevalence ratios (PRs) for exposed vs non-exposed adults and 95% confidence interval (CI). To account for the clustering effect of more adults selected from the same households, we adjusted for household level cluster effect using a resampling method. Then, we used a series of multivariate mixed effects Poisson regression models, considering various variables to estimate multivariate PR. Variables selections in each multivariate model was done by using bivariate models considering a P-value less than 0.1[36]. All models were adjusted for age and sex as general confounding variables and considered for household level cluster effect.

In Fig 1, a predefined conceptual framework that reflects our hypotheses about how risk factors for hypertension are clustered in blocks, the intricate relationship between these blocks and pathways through which these factors play role in hypertension. Statistical analysis was conducted based on a conceptual model. We performed a hierarchical effect decomposition (HED) approach to assess the impact of risk factors at various levels and to distinguish between the direct and medicated effects. The HED approach is detailed elsewhere [37,38].

Briefly, In model A, we use SES and general confounders like age and sex figure out how SES affects overall. Next in model B, we added educational attainment and in model C, we included block 2 variables; awareness and healthy behavior. It was followed by model D additionally including block 3 variable; body mass index (BMI) and model E additionally including block 4 variable; diabetes mellitus. When we looked at the impact of SES, we compared the PR of the lower SES vs richr in 2011 and 2018, while controlling for the next factor in the conceptual model. This gives an estimate of how factors along pathway have contributed to overall effect (Fig 1).

Furthermore, we computed multivariate attributable risks for exposed subjects (AR) and total study population (ART) according to formula:

$$AR = (PR - 1)/PR \times 100\% \qquad (1)$$

,and

$$ART = \frac{\sum_{i=1}^{n} P_i \times (PR_i - 1)}{1 + \sum_{i=1}^{n} P_i \times (PR_i - 1)} \times 100\% \qquad (2)$$

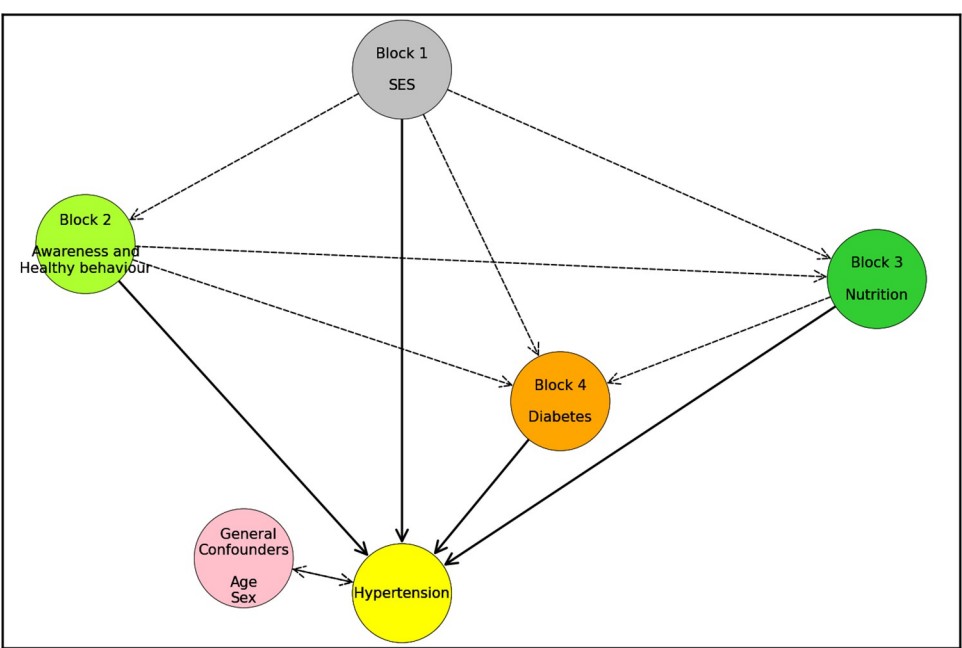

**Fig 1. Conceptual framework illustrating the inter-relationships between potential determinants (socio-economic status (SES), awareness, nutritional status, diabetes), and hypertension prevalence.** Age and sex are general confounders. The dash arrows indicate the attributable risk of the originating block (the arrow comes from) that is mediated by the receiving block (the arrow goes from). In contrast, the solid arrows represent the direct effect of blocks. The double arrow represents general confounders.

;where p is proportion of the population exposed [39–41]. Finally, we calculated mediating proportion (MP) by comparing the overall estimates of attributable risk of each factor with the estimates that were adjusted for SES according to the formula [16,35,37]:

$$MP = (ART_{unadj} - ART_{adj})/ART_{unadj} \times 100\%. \tag{3}$$

All statistical analysis was carried out using Python 3.10 and the statistical software package STATA version 13.1, (STATA Corporation, College Station, TX, USA).

## Results

The study population focused on adult aged 35 years and older in Bangladesh, as described elsewhere [25,32]. **Fig 2** shows the frequency, percentage and prevalence of hypertension, along with 95% confidence interval (95%CI) for each characteristics and overall estimated results from 2011 and 2018. A total of 7814 adults were analyzed in 2011. The average age of the study population was 51.3 (Standard Error (SE): 0.17 years). Males accounted for 3860 (49.4%), while 3787(48.4%) had no formal education. The prevalence of diabetes mellitus was 720 (9.6%), 840(10.8%) were awareness of their hypertension status, and the prevalence of excessive body weight was 1171(21.6%). Of the total participants, 956(17.6%) were overweight and 215(4.0%) were obese. The 8[th] survey in 2018 included 9015 adults for analysis with average age 45.5 years (SE: 0.19 years) for analysis. Males accounted for 5446 (60.4%), while 3017 (33.5%) had no formal education. The prevalence of diabetes mellitus was 857 (10.2%), 1295 (14.4%) were awareness of their hypertension status, and the prevalence of excessive body weight was 3422(38.5%). Of the total participants, 2514(28.3%) were overweight and 905

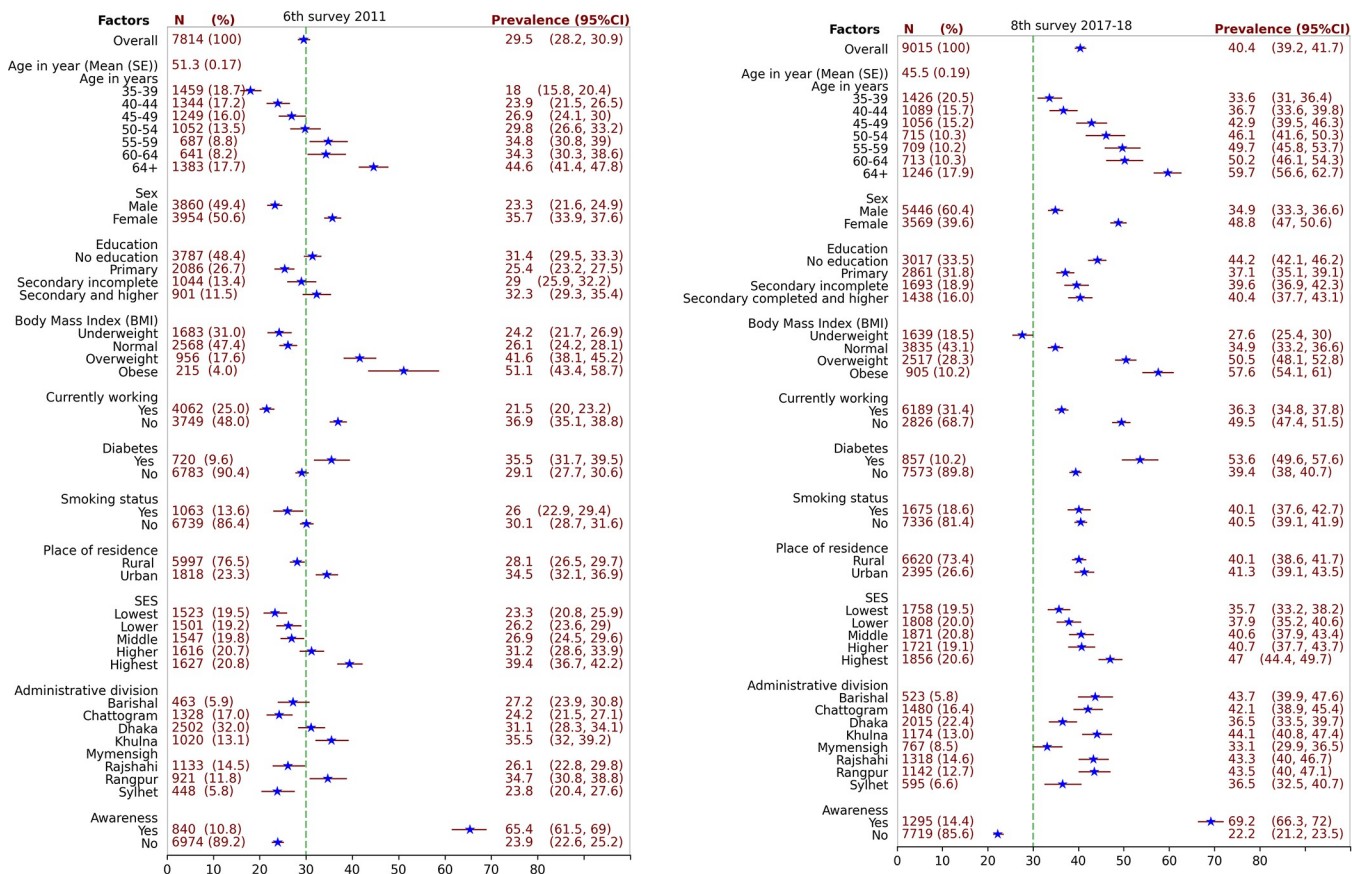

**Fig 2.** 6<sup>th</sup> survey 2011 (a) and 8<sup>th</sup> survey 2017–18 (b). The frequency, percentage and prevalence of hypertension along with associated confidence intervals (CI) for each characteristic and overall were estimated. The estimates were derived from the adult population using survey analysis with sampling weights. The green dash vertical line indicates the overall adult prevalence of hypertension at 6<sup>th</sup> survey in 2011.

(10.2%) were obese. The age-standard prevalence of hypertension increased by 10.9% from 29.5% (95%CI: 28.2–30.9) in 2011 to 40.4% (95%CI: 39.2–41.7) in 2018.

**Fig 3** presents the results of prevalence ratio (PR), along with 95% confidence intervals, decomposing the overall effect after adjusting for household level clustering and potential confounders (age and sex), along with upper level variables based on a conceptual framework depicted in Fig 1. However, it is important to note that mediating variables were not considered in each multivariate model. The findings from 6<sup>th</sup> survey in 2011 shows that, even after controlling for age, sex and place of residence, adults from richer people were more likely to have hypertension compared to those with lowest SES (middle SES: PR = 1.14, 95%CI: 1.01–1.29, higher SES: PR = 1.30, 95%CI: 1.16–1.47, highest: PR = 1.70, 95%CI: 1.52–1.90). Additionally, adults with secondary and higher education level exhibited a higher likelihood of hypertension (secondary and higher vs. no education: PR = 1.13, 95%CI: 1.01–1.28). Moreover, adults who were awareness of their hypertension status had a substantially higher likelihood of having hypertension (PR = 2.60, 95%CI: 2.38–2.83). Furthermore, excessive body weight was associated with an increased risk of hypertension compared to those with normal (underweight: PR = 0.81, 95%CI: 0.73–0.90, overweight: PR = 1.35, 95%CI: 1.22–1.49, and obese: PR = 1.44, 95%CI: 1.23–1.67). Similarly, the findings from 8<sup>th</sup> survey in 2018 shows that, even after controlling for age, sex and place of residence, adults from richer people were more likely to have hypertension compared to those with lowest SES (middle SES: PR = 1.16,

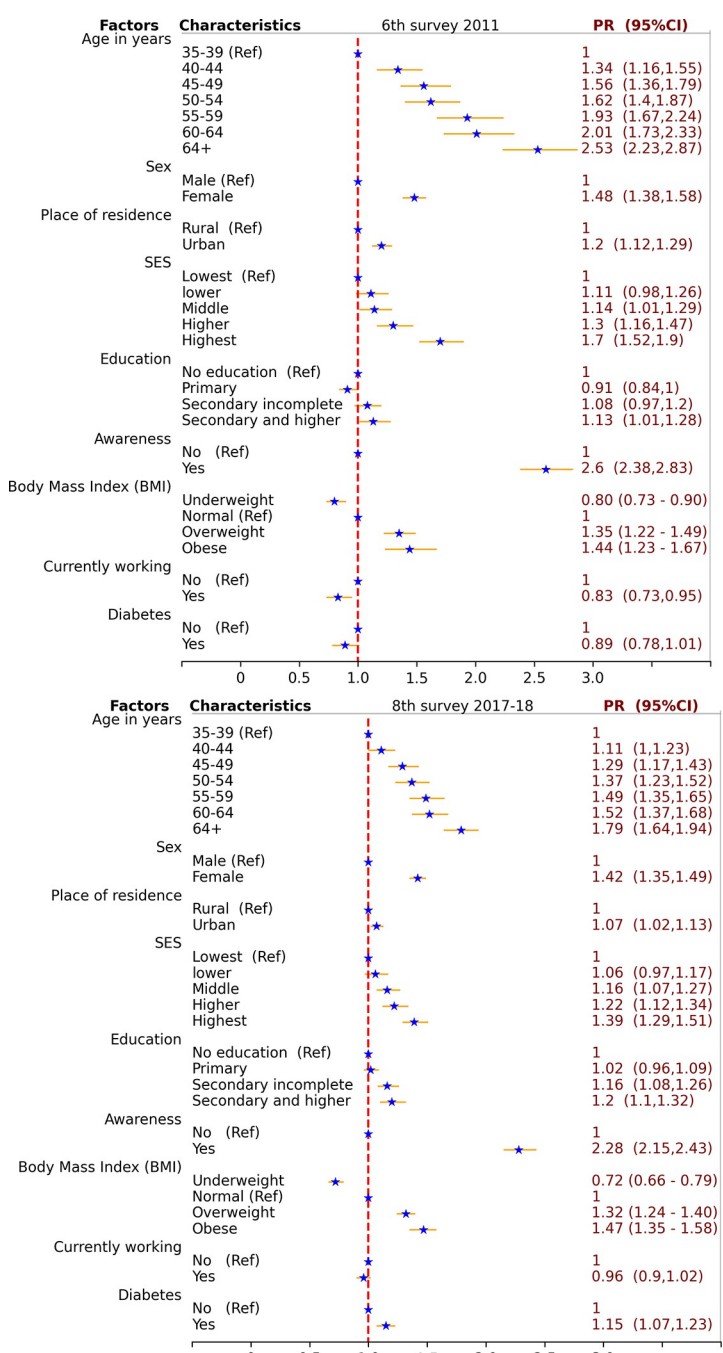

**Fig 3.** 6th survey 2011 (a), and 8th survey 2017–18 (b). Prevalence ratios (PRs) were estimated from mixed effect sequential multivariate Poisson regression models of hypertension based on conceptual framework depicted in Fig 1. The reference category had a PR of 1.0, meaning that it was no more likely to have hypertension. PRs greater than 1.0 indicate a positive association with hypertension, meaning that the category was more likely to have hypertension than the reference category. PRs less than 1.0 indicate a negative association with hypertension, meaning that the category was less likely to have hypertension than the reference category.

95%CI: 1.07–1.27, higher SES: PR = 1.22, 95%CI: 1.12–1.34, highest SES: PR = 1.39, 95%CI: 1.29–1.51). Additionally, adults with secondary and higher education level exhibited a higher likelihood of hypertension (secondary and higher vs. no education: PR = 1.20, 95%CI: 1.10–

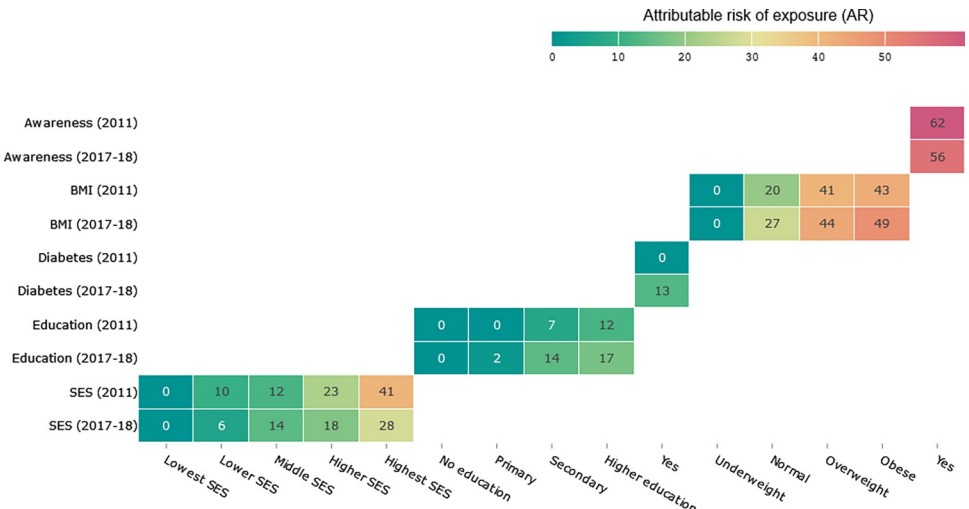

**Fig 4. Attributable risk of exposure (AR) was calculated for each category of exposure in the 6th survey 2011 and the 8th survey 2017–18.** The ARs were calculated using Formula (1), which is based on the multivariate estimates of relative risk (PR). The AR indicates the proportion of hypertension in exposed adults that can be attributed to the exposure.

1.32). Moreover, adults who were awareness of their hypertension status had a substantially higher likelihood of having hypertension (PR = 2.28, 95%CI: 2.15–2.43). Furthermore, excessive body weight was associated with an increased risk of hypertension compared to those with normal (underweight: PR = 0.72, 95%CI: 0.66–0.79, overweight: PR = 1.32, 95%CI: 1.24–1.40, and obese: PR = 1.47, 95%CI: 1.35–1.58).

In addition, **Fig 4** presents a comparison of the attributable risk (AR) for SES, education, BMI and awareness between the 6th survey in 2011 and 8th survey in 2018. The findings show an increase in AR across BMI over time. In the 6th survey in 2011, the ARs for SES were 10% for lower SES, 12% for middle SES, 23% for higher SES and 41% for highest SES. Education level had ARs of 12% for secondary and higher level, while hypertension awareness had an AR of 62%. For BMI, the ARs were 20% for normal, 41% for overweight and 43% for obesity. In the 8th survey in 2018, the ARs for SES were 6% for lower SES, 14% for middle SES, 18% for higher SES and 28% for highest SES. Education level had ARs of 17% for secondary and higher level, while hypertension awareness had an AR of 56%. For BMI, the ARs were 27% for normal, 44% for overweight and 49% for obesity.

**Fig 5** shows notable changes in the attributable risk in the total study population (ART) for SES, awareness and BMI from the 6th survey in 2011 to the 8th survey in 2017–18. Specifically, the ART for SES decreased from 21% in 2011 to 16% in 2018. Conversely, the ART for hypertension awareness showed a slight increase from 15% in 2011 to 16% in 2018. Notably, even after adjusted for SES, the ART for BMI increased from 24% in the 6th survey in 2011 to the 6th survey 36% in 2018. In addition, through our analytical approach we could explore if the increased hypertension awareness and improved nutrition status also altered the pathways through which the risk factors influenced the hypertension prevalence. We found that a significant portion of the influence of SES was mediated by other factors. The risk of hypertension that can be attributed to excessive body weight increases substantially. In 2011, the effect of higher SES was mediated by excessive body weight (51%) and awareness (39%). In contrast, in 2018, the effect of higher SES on hypertension was mediated by the different factors; 55% by excessive body weight and 41% by awareness.

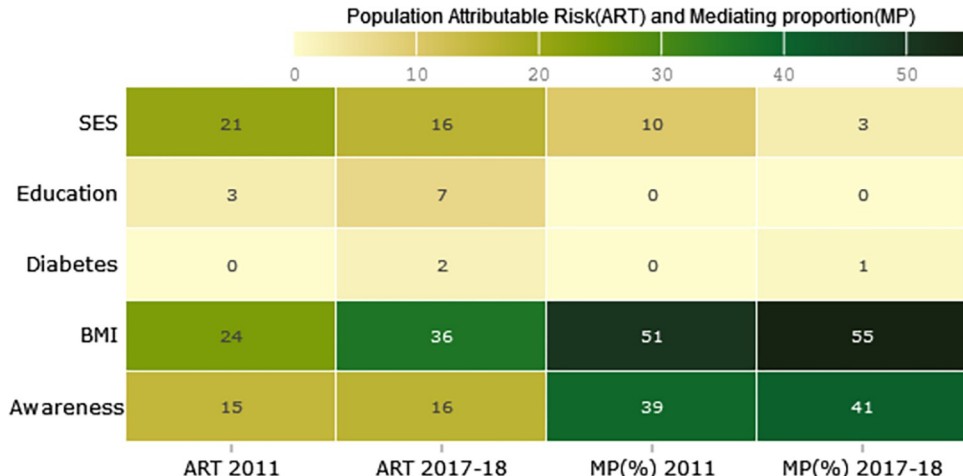

**Fig 5. The calculation of the total study population attributable risk (ART) for each factor using Formula (2) for both the 6th survey 2011 and the 8th survey 2017–18.** The estimate of the adjusted proportion of population attributable risk (ART) for hypertension due to exposure is determined by considering the proportion in unexposed subjects. Additionally, the mediating proportion (MP) for factors are calculated using Formula (3) for both surveys. The estimate of mediating proportion (MP) of each factor is derived from the adjusted $ART_{adj}$ and unadjusted $ART_{unadj}$ using Formula (3). MP indicates the percentage of the relationship between SES and hypertension that is mediated by these specific factors.

Fig 6 illustrates a reduction of ART of hypertension associated with SES by 5, while the distribution of SES remained relatively stable in the study population. However, increasing the educational level and excessive body weight led to substantially changes the population attributable risk of adult hypertension. Improving educational status explained an additional 4% of the hypertension risk in 2018. Furthermore, rising excessive body weight were observed to contribute to an additional 12% of the attributable risk of hypertension from 2011 to 2018.

## Discussion

This study identified the potential risk factors for hypertension in two surveys (2011 and 2018) and quantified relative, attributable and mediated risks by a HED modelling based on pre-defined conceptual framework. The most significant finding was the 10.9% increase in the

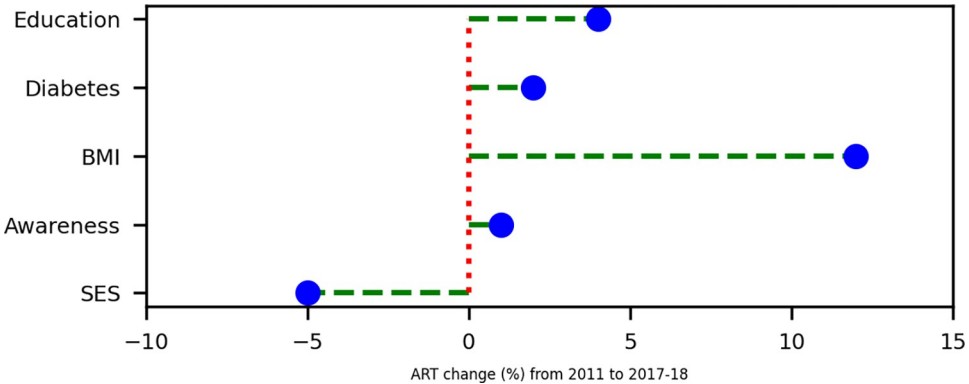

**Fig 6. Change in population attributable risk (ART) of the study population due to exposure to adult hypertension prevalence during 2011 and 2017–18, Bangladesh.** The proportion ART explained by mediating factors during 2011 and 2017–18. The proportion change of ART have been derived from 6th survey 2011 to 8th survey 2017–18 according to the formula: Change = ($ART2011 - ART_{2017-18}$).

hypertension prevalence over six-year period, indicating a growing burden of this condition on the public health. We also observed changes in the relative, attributable and mediated risks from 2011 to 2018. Specifically, we found that the effect of higher SES on adult hypertension was mostly mediated by self-awareness and excessive body weight.

It is surprising that the adult hypertension was increased by increasing exposure to proximal factors such as excessive body weight, and diabetes mellitus that are directly related to hypertension despite the Health Population and Nutrition Sector Development Program (HPNSDP) that implemented to increase population awareness of their hypertension status. For instance, by increasing unhealthy lifestyle such as smoking, higher education and urban population, remaining adult awareness of their hypertension status, and reducing underweighted population, the exposure to excessive body weight and diabetes mellitus increased from 2011 to 2018. As a result, the relative risk of unhealthy lifestyle, urban areas and higher SES decreased. However, the relative risk of excessive body weight, and diabetes mellitus increased. These findings suggest that the excessive body weight and diabetes mellitus were less closely related to poverty. In 2011, it was mainly the poor who did not consume dietary food, and had lack of awareness of hypertension, but other factors in 2018 including lack of awareness, higher education, less physical activity, unhealthy lifestyle and more urban population, are factors that likely contribute to strengthening the relationship between wealth and excessive body weight. Previous studies have shown a strong association between abnormal weight gain and hypertension because of imbalance in calories between energy consumption and spending [42–44]. Eating too many calories can lead to weight gain, as the excessive calories are stored as fat tissue in the body. This fat can also increase cholesterol levels in the blood, which can lead to high blood pressure [45].

Increased awareness of non-communicable diseases (NCD) from the 6th survey in 2011 to the 8th BDHS survey of 2017–18 has also reduced the relative risk of higher SES on hypertension. A wealthy family ability to consumption healthy food choices and increase in healthy behavior assumes greater importance in the prevalence of hypertension. As a consequence, before the national-wide awareness program in 2011, there was a strong positive association between higher socio-economic status (SES) and hypertension. This association became even stronger in 2018 than it was in 2011. Prior studies showed that the lower SES was strongly associated with hypertension in high income countries due to the impact of awareness and control of hypertension, inadequate education, poor lifestyles, and difficulty reaching a healthcare provider [18,46,47]. In contrast, this association is complicated and somewhat contradictory in low-income countries. The most rewarding finding from our study that the upper SES was strong positively associated with hypertension in 2011. After short time, this association were reduced in 2018. In 2011, higher SES was a major determinant of hypertension (attributable risk: 21). Most of the upper SES effect (90%) was mediated indicating that higher SES was closely related to other proximate hypertension factors such as excessive body weight, diabetes mellitus, and lack of hypertension awareness and control. A previous study in the same country revealed that these are potential risk factors [27,41]. In 2018, the situation had changed; higher SES had a less important role (attributable risk: 16%) but risk of hypertension was strongly related to excessive body weight (attributable risk: 36%) and awareness of hypertension status (attributable risk: 16%). These results suggest that the effect of awareness of hypertension status and excessive body weight on hypertension risk happened independently of SES. We also found that the effect of SES was mediated by excessive body weight, lack of awareness and diabetes mellitus. It relates to the awareness programs aimed at reducing health inequalities among different socioeconomic groups. [27,28,48].

The study's strengths are that it used data from two nationally representative surveys, which means that the results can be applied to other people, settings, and situations. Another strength

is that the field workers measured blood pressure, and anthropometric measurements using well-established techniques. In the multivariate analysis, mixed effect sequential Poisson regression models were used in cross-sectional survey to increase precision of the predictors. However, these cross-sectional surveys weakened our ability to deduce a cause-effect association. Furthermore, we defined hypertension based on blood pressure records at a single time point and awareness of hypertension status was self-reported. Moreover, a well-documented risk factors of hypertension including dietary food intake, salt intake, saturated fats, physical activities, sleeping habit, smoking, and alcohol consumption data were not collected, which was not possible to adjust for in our comparison analysis. Additionally, our comparative analysis carried out for two different sets of populations. We couldn't eliminate the possibility of biases stemming from unobserved variables.

## Conclusion

This study shows that the prevalence of hypertension in Bangladesh has increased significantly over the years. Several potential determinants of adult hypertension were identified, including socio-economic status, awareness of hypertension, and excessive body weight. Additionally, the rising prevalence of other non-communicable diseases, such as diabetes mellitus and excessive body weight, has exacerbated the relative, attributable, and mediating risks associated with hypertension. This has changed the causal pathway between these determinants and adult hypertension. Well-designed interventions can mitigate adult hypertension and related health challenges in Bangladesh. We must prioritize public health strategies that promote awareness, healthy lifestyles, and equitable access to healthcare.Bottom of Form

## Acknowledgments

The authors would like to acknowledge the program of demographic and health survey (DHS) team for providing survey data. The author(s) would like to acknowledge the contribution of caregivers/ mothers of the study participants for their consent to enroll children in the study. icddr,b is also grateful to the Governments of Bangladesh, Canada, for providing core/unrestricted support.

## Author Contributions

**Conceptualization:** Probir Kumar Ghosh, Md. Golam Dostogir Harun, Ireen Sultana Shanta, Ausraful Islam, Kaniz Khatun E. Jannat, Haider Mannan.

**Data curation:** Probir Kumar Ghosh.

**Formal analysis:** Probir Kumar Ghosh, Md. Golam Dostogir Harun, Ireen Sultana Shanta, Ausraful Islam, Kaniz Khatun E. Jannat, Haider Mannan.

**Investigation:** Probir Kumar Ghosh, Md. Golam Dostogir Harun, Haider Mannan.

**Methodology:** Probir Kumar Ghosh, Md. Golam Dostogir Harun, Ireen Sultana Shanta, Ausraful Islam, Kaniz Khatun E. Jannat, Haider Mannan.

**Resources:** Probir Kumar Ghosh, Haider Mannan.

**Software:** Probir Kumar Ghosh, Haider Mannan.

**Supervision:** Probir Kumar Ghosh, Kaniz Khatun E. Jannat, Haider Mannan.

**Visualization:** Probir Kumar Ghosh, Haider Mannan.

**Writing – original draft:** Probir Kumar Ghosh, Md. Golam Dostogir Harun, Ireen Sultana Shanta, Ausraful Islam, Kaniz Khatun E. Jannat, Haider Mannan.

**Writing – review & editing:** Probir Kumar Ghosh, Md. Golam Dostogir Harun, Ireen Sultana Shanta, Ausraful Islam, Kaniz Khatun E. Jannat.

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
