## [Decision Letter · Decision Letter 0]

17 Sep 2023

PONE-D-23-25419Prevalence and determinants of hypertension among older adults: A comparative analysis of the 6th and 8th national health surveys of BangladeshPLOS ONE

Dear Dr. Ghosh,

Thank you for submitting your manuscript to PLOS ONE. After careful consideration, we feel that it has merit but does not fully meet PLOS ONE’s publication criteria as it currently stands. Therefore, we invite you tosubmit a revised version of the manuscript that addresses the points raised during the review process. Your manuscript has been reviewed and the comments are below this letter. Address all the comments below and resubmit. Please submit your revised manuscript by Nov 01 2023 11:59PM. If you will need more time than this to complete your revisions, please reply to this message or contact the journal office at plosone@plos.org. Please include the following items when submitting your revised manuscript:A rebuttal letter that responds to each point raised by the academic editor and reviewer(s). You should upload this letter as a separate file labeled 'Response to Reviewers'.A marked-up copy of your manuscript that highlights changes made to the original version. You should upload this as a separate file labeled 'Revised Manuscript with Track Changes'.An unmarked version of your revised paper without tracked changes. You should upload this as a separate file labeled 'Manuscript'.If applicable, we recommend that you deposit your laboratory protocols in protocols.io to enhance the reproducibility of your results. Protocols.io assigns your protocol its own identifier (DOI) so that it can be cited independently in the future. For instructions see: https://journals.plos.org/plosone/s/submission-guidelines#loc-laboratory-protocols. Additionally, PLOS ONE offers an option for publishing peer-reviewed Lab Protocol articles, which describe protocols hosted on protocols.io. Read more information on sharing protocols at https://plos.org/protocols?utm_medium=editorial-email&utm_source=authorletters&utm_campaign=protocols.

We look forward to receiving your revised manuscript.

Kind regards,

Mpho Keetile, PhD

Academic Editor

PLOS ONE

Journal Requirements:

2. Thank you for submitting the above manuscript to PLOS ONE. During our internal evaluation of the manuscript, we found significant text overlap between your submission and previous work in the [introduction, conclusion, etc.].

Please revise the manuscript to rephrase the duplicated text, cite your sources, and provide details as to how the current manuscript advances on previous work. Please note that further consideration is dependent on the submission of a manuscript that addresses these concerns about the overlap in text with published work.

[If the overlap is with the authors’ own works: Moreover, upon submission, authors must confirm that the manuscript, or any related manuscript, is not currently under consideration or accepted elsewhere. If related work has been submitted to PLOS ONE or elsewhere, authors must include a copy with the submitted article. Reviewers will be asked to comment on the overlap between related submissions (http://journals.plos.org/plosone/s/submission-guidelines#loc-related-manuscripts).]

We will carefully review your manuscript upon resubmission and further consideration of the manuscript is dependent on the text overlap being addressed in full. Please ensure that your revision is thorough as failure to address the concerns to our satisfaction may result in your submission not being considered further.

"No.The funders had no role in study design, data collection and analysis, decision to publish, or preparation of the manuscript." 

6. review your reference list to ensure that it is complete and correct. If you have cited papers that have been retracted, please include the rationale for doing so in the manuscript text, or remove these references and replace them with relevant current references. Any changes to the reference list should be mentioned in the rebuttal letter that accompanies your revised manuscript. If you need to cite a retracted article, indicate the article’s retracted status in the References list and also include a citation and full reference for the retraction notice.

Additional Editor Comments:

The reviewers have provided feed back for the manuscript. The manuscript is well written. However there are some minor comments attached that you need to address before the manuscript can be accepted. Make sure that all references cited in the document are in the reference list and vice versa. Correct all typos and grammatical errors

Reviewers' comments:

Reviewer's Responses to Questions

**Comments to the Author**

1. Is the manuscript technically sound, and do the data support the conclusions?

Reviewer #1: Yes

Reviewer #2: Yes

2. Has the statistical analysis been performed appropriately and rigorously? 

Reviewer #1: Yes

Reviewer #2: I Don't Know

3. Have the authors made all data underlying the findings in their manuscript fully available?

Reviewer #1: Yes

Reviewer #2: Yes

4. Is the manuscript presented in an intelligible fashion and written in standard English?

Reviewer #1: Yes

Reviewer #2: Yes

5. Review Comments to the Author

Reviewer #1: This is a good manuscript. However, 2 points need to be addressed. They are as outlined below:

1) The limitations of this study must be outlined.

2) The utility or uniqueness of this work must be highlighted.

Reviewer #2: Prevalence and determinants of hypertension among older adults: A comparative analysis of the 6th and 8th national health surveys of Bangladesh

Manuscript Review:

General:

The rationale is straight forward. The objectives of the study are clear. For the most part the methods are appropriate and well described. The author further summarized the strength and limitation of the study. Despite these strength, there are some aspects of the manuscript that needs to be addressed including several lapses in English construction throughout the manuscript, which makes it difficult to read sometimes. The authors should consider having their manuscript proofread before publication.

Abstract

This is well written and contains all the elements of a good abstract, but there are minor corrections to make:

1. Line 21: add a word “and” in the sentence “… in low-and middle income…”

2. The second sentence in background section is not clear, so the author need to rephrase this sentence.

3. Line 27: add a word “data” for clarity in sentence “……..were assessed in 2011 and 2018 data of Bangladesh….”

4. Line 28: Rephrase the sentence i.e “…enrolled both men and women aged above 34 years and….”

5. Line 37: The sentence is not clear to the reader, the author need to rephrase “Socioeconomic status (SES) was a major distal determinant of hypertension and had 21% population attributable risk…”

6. Line 40: The sentence is no clear, please rephrase i.e “……, with 97% of risk mediated by excessive….” Also the author should change the word “aware” to “awareness”

Introduction:

This is also well written, except for minor corrections

1. Line 58: add a word “and” in the sentence “… in low-and middle income…” similarly to Line 60

2. Line 93: change “measure of” to “measurement of”

Methods:

By and large this section is okay, except for minor comments:

1. Some sentences are not clear to the reader and therefore need to be rephrased for the reader to easily follow and understand, for-instance Line 113, 114, 123 and 129:

• Line 113: Suggested “of three adults aged more than 34 years”

• Line 114: Suggested “In order to have more focus on non-communicable diseases, the 4th HPNSP 2016-2021 included a component….”

• Line 123: Suggested “The survey involved 18,000 and 20,160 households obtained by using two-stage stratified cluster sampling method in 2011 and 2017-18 respectively”

• Line 129: Suggested “we used data of the 6th and 8th Bangladesh Demographic and Health surveys (BDHSs) conducted from March to October 2011 and from October 2017 to March 2018 respectively”

• Line 237: Suggested “We classified an individual as hypertensive if ........”

2. The author should specify the devices used to measure blood pressure by briefly describing the Model Company and country of origin.

3. The author did not tell how many times Blood pressure was measured and whether was on different occasions as recommended for diagnosis of hypertension. This need to come out clearly.

4. The author should give a brief description of how weight and weight was measured and tools used or can provide a reference of published paper for that work which have these description in the methodology.

5. Author need to provide reference for the BMI cut-off point used.

Results

This section is okay except for minor comments:

1. The author should state clearly the diabetes being referred in this study to be sure is not the other one (Diabetes insipidus vs Diabetes mellitus) i.e Diabetes mellitus (DM)

2. The author should correct grammar error and missing word noted in line 188, 195, 227, 239,

• Line 188: add word “years” suggested “The average age of the study population was 51.3 (SE: 0.17 years)” and SE should be written in long for in the first place with abbreviation in brackets and continue to use abbreviation in subsequent use.

• Line 195: Remove the word “and” after the word “prevalence of”

• Line 227: Suggested to use the word “and” instead of “to”

• Line 239: A word “slightly” should be replaced by “slight” and a word “increased” replaced by “increase”

• Line 240 – 241: add the word “in” before the word “the 6th survey...”

• Line 242: suggested to replace the word “increasing” with “increased”, remove the word “the” before the word “awareness” and before “nutrition” and change the word “improving to “improved”

• Line 252: add the word “and” and remove comma between the word “education level” and “excessive body weight”

3. Line 191 – 192: I think here the percent should be calculated out of those with excessive body weight but as it stands seems to be calculated out of total. Similarly to line 196 - 197

4. Line 192 - 193: The author advised to combine the two sentences. Suggested “The 8th survey in 2018 included 9015 adults with average age of 45.5 (SE: 0.19) for analysis.

5. Line 230,234 239 and 242: Add the word “hypertension” before the word “awareness”

6. Line 244: The sentence “ The proportion of attributable……….” Is not clear. Please rephrase

Discussion and conclusion

The discussion and conclusions are by and large in line with the findings. The authors have discussed their results well and the conclusions are appropriate. However, there some minor comments to address for better flow and easy for a reader to follow.

1. The author should correct grammar error and missing word noted in line 261, 268, 273, 286, 300, 309, 312 and 316

• Line 261: replace “growth” with “growing”

• Line 268 and 316: Replace the word “aware” with “awareness”

• Line 273: Add the word “and” between “education level” and “excessive body weight”

• Line 281: Add the word “the” before the word “body”

• Line 286: Replace the word “intake” with consumption

• Line 290: remove “a” before the word strongly

• Line 300: Replace “controlling” with “control”

• Line 309: Rephrase the sentence

• Line 312: Add the word “and” between words “blood pressure” and anthropometric parameters”

2. Line 271: The author is advised to break this sentence “As a result, the relative risk of unhealthy lifestyle, urban areas and higher SES decreased……..” into two for clarity to the reader.

3. Line 282: The author should give a summary of a link between high blood cholesterol level and development of high blood pressure.

4. Line 292 – 293: The author is advised to avoid the use of the word “lack” and instead use the word “poor” or “inadequate”. Also the statement “lack of access to health seeking facilities” is not clear, Please check and correct.

5. Line 294 – 296: The sentence “The most rewarding finding from our study that the upper SES was strong positively…….” Is very long and not clear to the reader. The author is advised to rephrase and break the sentence.

6. PLOS authors have the option to publish the peer review history of their article (what does this mean?). If published, this will include your full peer review and any attached files.

Reviewer #1: **Yes: **KAUSHIK BOSE

Reviewer #2: No

---

## [Author Response · Author response to Decision Letter 0]

22 Sep 2023

Response to reviewers:

Reviewer #1: This is a good manuscript. However, 2 points need to be addressed. They are as outlined below:

1) The limitations of this study must be outlined.

2) The utility or uniqueness of this work must be highlighted.

Autor’s responses: Thanks for pointed out this issue. Authors are grateful for raising these issues. The limitation and strength have been revised.

Reviewer #2: Prevalence and determinants of hypertension among older adults: A comparative analysis of the 6th and 8th national health surveys of Bangladesh

Manuscript Review:

General:

The rationale is straight forward. The objectives of the study are clear. For the most part the methods are appropriate and well described. The author further summarized the strength and limitation of the study. Despite these strength, there are some aspects of the manuscript that needs to be addressed including several lapses in English construction throughout the manuscript, which makes it difficult to read sometimes. The authors should consider having their manuscript proofread before publication.

Autor’s responses: Thanks for your comments. Authors appreciate your comments and have careful revised according to reviewer’s suggestions.

Abstract

This is well written and contains all the elements of a good abstract, but there are minor corrections to make:

1. Line 21: add a word “and” in the sentence “… in low-and middle income…”

Autor’s responses: Thanks for your comments. Authors have careful revised according to reviewer’s suggestions. Modified texts are in red color.

Thank you for your comments. We have carefully revised the manuscript according to your suggestions.

2. The second sentence in background section is not clear, so the author need to rephrase this sentence.

Autor’s responses: Thanks for your comments. Authors have careful revised according to reviewer’s suggestions. Modified texts are in red color.

3. Line 27: add a word “data” for clarity in sentence “……..were assessed in 2011 and 2018 data of Bangladesh….”

Autor’s responses: Thanks for your comments.Authors careful revision has been conducted as per comments. Modified texts are in red color.

4. Line 28: Rephrase the sentence i.e “…enrolled both men and women aged above 34 years and….”

Autor’s responses: Thanks for your comments. Authors have careful revised according to reviewer’s suggestions. Modified texts are in red color.

5. Line 37: The sentence is not clear to the reader, the author need to rephrase “Socioeconomic status (SES) was a major distal determinant of hypertension and had 21% population attributable risk…”

Autor’s responses: Thanks for your comments. Authors have careful revised according to reviewer’s suggestions. Modified texts are in red color.

6. Line 40: The sentence is no clear, please rephrase i.e “……, with 97% of risk mediated by excessive….” Also the author should change the word “aware” to “awareness” 

Autor’s responses: Thanks for your comments. Authors have careful revised according to reviewer’s suggestions. Modified texts are in red color.

Introduction:

This is also well written, except for minor corrections

1. Line 58: add a word “and” in the sentence “… in low-and middle income…” similarly to Line 60

Autor’s responses: Thanks for your comments. Authors have careful revised according to reviewer’s suggestions. Modified texts are in red color. 

2. Line 93: change “measure of” to “measurement of” 

Autor’s responses: Thanks for your comments. Authors have careful revised according to reviewer’s suggestions. Modified texts are in red color.

Methods:

By and large this section is okay, except for minor comments:

1. Some sentences are not clear to the reader and therefore need to be rephrased for the reader to easily follow and understand, for-instance Line 113, 114, 123 and 129:

Autor’s responses: Thanks for your comments. Authors have careful revised according to reviewer’s suggestions. Modified texts are in red color.

• Line 113: Suggested “of three adults aged more than 34 years”

Autor’s responses: Thanks for suggestion. Authors have careful revised according to reviewer’s suggestions. Modified texts are in red color.

• Line 114: Suggested “In order to have more focus on non-communicable diseases, the 4th HPNSP 2016-2021 included a component….” 

Autor’s responses: Thanks for suggestion. Authors have careful revised according to reviewer’s suggestions. Modified texts are in red color.

• Line 123: Suggested “The survey involved 18,000 and 20,160 households obtained by using two-stage stratified cluster sampling method in 2011 and 2017-18 respectively”

Autor’s responses: Thanks for suggestion. Authors have careful revised according to reviewer’s suggestions. Modified texts are in red color.

• Line 129: Suggested “we used data of the 6th and 8th Bangladesh Demographic and Health surveys (BDHSs) conducted from March to October 2011 and from October 2017 to March 2018 respectively”

Autor’s responses: Thanks for suggestion. Authors have careful revised according to reviewer’s suggestions. Modified texts are in red color.

• Line 237: Suggested “We classified an individual as hypertensive if ........”

Autor’s responses: Thanks for suggestion. Authors have careful revised according to reviewer’s suggestions. Modified texts are in red color.

2. The author should specify the devices used to measure blood pressure by briefly describing the Model Company and country of origin.

Autor’s responses: Thanks for suggestion. Authors have careful revised according to reviewer’s suggestions. Modified texts are in red color.

3. The author did not tell how many times Blood pressure was measured and whether was on different occasions as recommended for diagnosis of hypertension. This need to come out clearly.

Autor’s responses: Thanks for suggestion. Authors have careful revised according to reviewer’s suggestions. Modified texts are in red color. 

4. The author should give a brief description of how weight and weight was measured and tools used or can provide a reference of published paper for that work which have these description in the methodology.

Autor’s responses: Thanks for suggestion. Authors have careful revised according to reviewer’s suggestions. Modified texts are in red color. 

5. Author need to provide reference for the BMI cut-off point used.

Autor’s responses: Thanks for suggestion. Authors have careful revised according to reviewer’s suggestions. Modified texts are in red color. 

Results

This section is okay except for minor comments:

1. The author should state clearly the diabetes being referred in this study to be sure is not the other one (Diabetes insipidus vs Diabetes mellitus) i.e Diabetes mellitus (DM)

Autor’s responses: Thanks for suggestion. Authors have careful revised according to reviewer’s suggestions. Modified texts are in red color.

2. The author should correct grammar error and missing word noted in line 188, 195, 227, 239,

Autor’s responses: Authors appreciate your suggestion and Authors have careful revised according to reviewer’s suggestions. Modified texts are in red color.

• Line 188: add word “years” suggested “The average age of the study population was 51.3 (SE: 0.17 years)” and SE should be written in long for in the first place with abbreviation in brackets and continue to use abbreviation in subsequent use.

Autor’s responses: Thanks for suggestion. Authors have careful revised according to reviewer’s suggestions. Modified texts are in red color. 

• Line 195: Remove the word “and” after the word “prevalence of”

Autor’s responses: Thanks for suggestion. Authors have careful revised according to reviewer’s suggestions. Modified texts are in red color. 

• Line 227: Suggested to use the word “and” instead of “to”

Autor’s responses: Thanks for suggestion. Authors have careful revised according to reviewer’s suggestions. Modified texts are in red color. 

• Line 239: A word “slightly” should be replaced by “slight” and a word “increased” replaced by “increase”

Autor’s responses: Thanks for suggestion. Authors have careful revised according to reviewer’s suggestions. Modified texts are in red color. 

• Line 240 – 241: add the word “in” before the word “the 6th survey...”

Autor’s responses: Thanks for suggestion. Authors have careful revised according to reviewer’s suggestions. Modified texts are in red color. 

• Line 242: suggested to replace the word “increasing” with “increased”, remove the word “the” before the word “awareness” and before “nutrition” and change the word “improving to “improved”

Autor’s responses: Thanks for suggestion. Authors have careful revised according to reviewer’s suggestions. Modified texts are in red color. 

• Line 252: add the word “and” and remove comma between the word “education level” and “excessive body weight” 

Autor’s responses: Thanks for suggestion. Authors have careful revised according to reviewer’s suggestions. Modified texts are in red color. 

3. Line 191 – 192: I think here the percent should be calculated out of those with excessive body weight but as it stands seems to be calculated out of total. Similarly to line 196 – 197

Autor’s responses: Thanks for suggestion. Authors have careful revised according to reviewer’s suggestions. Modified texts are in red color. 

4. Line 192 - 193: The author advised to combine the two sentences. Suggested “The 8th survey in 2018 included 9015 adults with average age of 45.5 (SE: 0.19) for analysis.

Autor’s responses: Thanks for suggestion. Authors have careful revised according to reviewer’s suggestions. Modified texts are in red color. 

5. Line 230,234 239 and 242: Add the word “hypertension” before the word “awareness”

Autor’s responses: Thanks for suggestion. Authors have careful revised according to reviewer’s suggestions. Modified texts are in red color.

6. Line 244: The sentence “ The proportion of attributable……….” Is not clear. Please rephrase

Autor’s responses: Thanks for suggestion. Authors have careful revised according to reviewer’s suggestions. Modified texts are in red color. 

Discussion and conclusion

The discussion and conclusions are by and large in line with the findings. The authors have discussed their results well and the conclusions are appropriate. However, there some minor comments to address for better flow and easy for a reader to follow.

1. The author should correct grammar error and missing word noted in line 261, 268, 273, 286, 300, 309, 312 and 316

Autor’s responses: Authors appreciate your suggestion and Authors have careful revised according to reviewer’s suggestions. Modified texts are in red color.

• Line 261: replace “growth” with “growing”

Autor’s responses: Thanks for suggestion. Authors have careful revised according to reviewer’s suggestions. Modified texts are in red color.

• Line 268 and 316: Replace the word “aware” with “awareness”

Autor’s responses: Authors appreciate your suggestion and Authors have careful revised according to reviewer’s suggestions. Modified texts are in red color.

• Line 273: Add the word “and” between “education level” and “excessive body weight”

Autor’s responses: Thanks for suggestion. Authors have careful revised according to reviewer’s suggestions. Modified texts are in red color.

 • Line 281: Add the word “the” before the word “body”

Autor’s responses: Thanks for suggestion. Authors have careful revised according to reviewer’s suggestions. Modified texts are in red color.

• Line 286: Replace the word “intake” with consumption

Autor’s responses: Thanks for suggestion. Authors have careful revised according to reviewer’s suggestions. Modified texts are in red color.

• Line 290: remove “a” before the word strongly

Autor’s responses: Thanks. Authors are grateful for raising this issue. Authors have careful revised according to reviewer’s suggestions. Modified texts are in red color.

• Line 300: Replace “controlling” with “control”

Autor’s responses: Thanks. Authors have careful revised according to reviewer’s suggestions. Modified texts are in red color.

• Line 309: Rephrase the sentence

Autor’s responses: Thanks. Authors have careful revised according to reviewer’s suggestions. Modified texts are in red color.

• Line 312: Add the word “and” between words “blood pressure” and anthropometric parameters”

Autor’s responses: Thanks. Authors have careful revised according to reviewer’s suggestions. Modified texts are in red color.

2. Line 271: The author is advised to break this sentence “As a result, the relative risk of unhealthy lifestyle, urban areas and higher SES decreased……..” into two for clarity to the reader.

Autor’s responses: Thanks. Authors have careful revised according to reviewer’s suggestions. Modified texts are in red color.

3. Line 282: The author should give a summary of a link between high blood cholesterol level and development of high blood pressure.

Autor’s responses: Thanks. Authors are grateful for raising this issue. Authors have careful revised according to reviewer’s suggestions. Modified texts are in red color.

4. Line 292 – 293: The author is advised to avoid the use of the word “lack” and instead use the word “poor” or “inadequate”. Also the statement “lack of access to health seeking facilities” is not clear, Please check and correct.

Autor’s responses: Authors appreciate your comment. Authors have careful revised according to reviewer’s suggestions. Modified texts are in red color. 

5. Line 294 – 296: The sentence “The most rewarding finding from our study that the upper SES was strong positively…….” Is very long and not clear to the reader. The author is advised to rephrase and break the sentence.

Autor’s responses: Thanks, and appreciate your suggestion. Authors have careful revised according to reviewer’s suggestions. Modified texts are in red color.

Response to Editor:

Thanks, and appreciate your suggestion. Authors have careful revised according to editor's suggestions and comments.

---

## [Editor Report · Decision Letter 1]

4 Oct 2023

Prevalence and determinants of hypertension among older adults: A comparative analysis of the 6th and 8th national health surveys of Bangladesh

PONE-D-23-25419R1

Dear Dr. Probir Kumar Ghosh,

We’re pleased to inform you that your manuscript has been judged scientifically suitable for publication and will be formally accepted for publication once it meets all outstanding technical requirements.

Kind regards,

Mpho Keetile, PhD

Academic Editor

PLOS ONE
---

## [Editor Report · Acceptance letter]

6 Oct 2023

PONE-D-23-25419R1 

Prevalence and determinants of hypertension among older adults: A comparative analysis of the 6^th^ and 8^th^ national health surveys of Bangladesh 

Dear Dr. Ghosh:

I'm pleased to inform you that your manuscript has been deemed suitable for publication in PLOS ONE. Congratulations! Your manuscript is now with our production department. 

Kind regards, 

on behalf of

Dr. Mpho Keetile 

Academic Editor

PLOS ONE